



# Urban aerosol chemistry at a land-water transition site during summer - Part 1: Impact of agricultural and industrial ammonia emissions

Nicholas Balasus[1], Michael A. Battaglia[1], Katherine Ball[1], Vanessa Caicedo[2], Ruben Delgado[2], Annmarie G. Carlton[3], and Christopher J. Hennigan[1]

[1]Department of Chemical, Biochemical, and Environmental Engineering, University of Maryland, Baltimore County, Baltimore, MD, 21250

[2]Joint Center for Earth Systems Technology, University of Maryland, Baltimore County, Baltimore, MD, 21250

[3]Department of Chemistry, University of California, Irvine, CA 92697, USA

**Correspondence:** Christopher J. Hennigan (hennigan@umbc.edu)

**Abstract.** This study characterizes the impact of the Chesapeake Bay and associated meteorological phenomena on aerosol chemistry during the second Ozone Water-Land Environmental Transition Study (OWLETS-2) field campaign during summer 2018. Measurements of inorganic $PM_{2.5}$ composition, gas-phase ammonia ($NH_3$), and an array of meteorological parameters were undertaken at Hart-Miller Island (HMI), a land-water transition site just east of downtown Baltimore on the Chesapeake

Bay. The observations at HMI were characterized by abnormally high $NH_3$ concentrations (maximum of 19.3 $\mu g\,m^{-3}$, average of 3.83 $\mu g\,m^{-3}$), which were more than a factor of three higher than $NH_3$ levels measured at the closest Atmospheric Ammonia Network (AMoN) site (approximately 45 km away). While sulfate concentrations at HMI agreed quite well with those measured at a regulatory monitoring station 45 km away, aerosol ammonium and nitrate concentrations were significantly higher, due to the ammonia-rich conditions that resulted from the elevated $NH_3$. The high $NH_3$ concentrations were

largely due to regional agricultural emissions, including dairy farms in southeastern Pennsylvania and poultry operations in the Delmarva Peninsula (Delaware-Maryland-Virginia). Reduced $NH_3$ deposition during transport over the Chesapeake Bay likely contributed to enhanced concentrations at HMI compared to the more inland AMoN site. Several peak $NH_3$ events were recorded, including the maximum $NH_3$ observed during OWLETS-2, that appear to originate from a cluster of industrial sources near downtown Baltimore. Such events were all associated with nighttime emissions and advection to HMI under low

wind speeds ($< 1\,m\,s^{-1}$) and stable atmospheric conditions. Our results demonstrate the importance of industrial sources, including several that are not represented in the emissions inventory, on urban air quality. Together with our companion paper, which examines aerosol liquid water and pH during OWLETS-2, we highlight unique processes affecting urban air quality of coastal cities that are distinct from continental locations.

## 1 Introduction

Inorganic salts, especially sulfate, nitrate, and ammonium, contribute a significant fraction of fine particulate matter ($PM_{2.5}$) mass in the atmosphere. Ammonia ($NH_3$) is an important component in this system due to its reaction with acidic species,





especially $H_2SO_4$ (or $HSO_4^-$) and $HNO_3$, to form secondary $PM_{2.5}$. Volatility differences between $H_2SO_4$, $HNO_3$, and $NH_3$ result in a highly non-linear system that is strongly influenced by the aerosol liquid water content, and by small concentrations of nonvolatile aerosol species (Seinfeld and Pandis, 2016; Guo et al., 2018). The impact of $NH_3$ on inorganic particulate

matter formation has been extensively studied (e.g., Adams et al., 1999; Ansari and Pandis, 1998; Pinder et al., 2008; Bian et al., 2017). A common finding among these studies, and others, is that the $NH_3$-$H_2SO_4$-$HNO_3$-$H_2O$ system is characterized by "ammonia rich" and "ammonia poor" chemical regimes that produce substantial differences in aerosol composition and mass. These differences have direct implications for aerosol effects on public health (Pozzer et al., 2017; Paulot and Jacob, 2014), nutrient deposition and ecosystem health (Nenes et al., 2020; Bergström and Jansson, 2006), and global climate change

(Turnock et al., 2019).

Emissions of the secondary aerosol precursors $SO_2$, $NO_x$, and $NH_3$ directly affect regional $PM_{2.5}$ concentrations (Pinder et al., 2007; Zhao et al., 2013). Air quality regulations in the US have substantially decreased $SO_2$ and $NO_x$ emissions, leading to decreases in ambient $PM_{2.5}$ (Hand et al., 2012a), while $NH_3$ has remained largely unregulated. Reductions of $NH_3$ emissions have been suggested as a cost-effective way to reduce $PM_{2.5}$, as well (Pinder et al., 2007; Backes et al., 2016). Observations

of wet deposition suggest $NH_3$ emissions have actually been rising over the past few decades (Keene et al., 2014). Globally, agricultural activity (from livestock and fertilizer application to crops) is the dominant source of $NH_3$ emissions (Paulot et al., 2015; Pinder et al., 2006; Sutton et al., 2013). Synthetic fertilizer production has recently been identified as an intense local $NH_3$ source (Van Damme et al., 2018), though its contribution to global emissions is likely small compared to volatilization from synthetic fertilizer applications (Bouwman et al., 1997).

While agriculture collectively emits a dominant fraction of $NH_3$ globally, numerous other sources contribute to local or regional $NH_3$ levels, especially in urban areas that are not in close proximity to major agricultural operations. Motor vehicles equipped with catalytic converters can reduce NO to $NH_3$ in an undesired side reaction, resulting in significant $NH_3$ emissions (Perrino et al., 2002; Sun et al., 2017; Kean et al., 2009). Anaerobic protein degradation produces $NH_3$, as well, resulting in substantial atmospheric emissions from anthropogenic sources present in all urban areas, including landfills, composting, and

wastewater treatment (Rittmann and McCarty, 2001; Artíñano et al., 2018). Numerous industrial sources emit $NH_3$ (Meng et al., 2017), while natural sources (e.g., natural vegetation, wild animals, and large bodies of water) contribute to background $NH_3$ levels (Paulot et al., 2015; Sutton et al., 2013). Biomass burning also emits $NH_3$, though the effects are often seasonal and are most pronounced in regions with intense burning activities (Paulot et al., 2017). In the eastern US, biomass burning intensity is higher in spring and fall than in the summer (Washenfelder et al., 2015).

Urban air quality is controlled by emissions from multiple sectors, meteorology, and dynamic chemical processes that evolve throughout the day. Coastal cities often experience contributions from unique sources (e.g., ocean and shipping emissions), with chemical processes in the polluted marine boundary layer that are distinct from other regions in the atmosphere (e.g., de Gouw et al., 2005; Vutukuru and Dabdub, 2008). Further, meteorological processes associated with the land-water transition can exert a controlling effect on urban air quality that are absent in continental cities. One such phenomenon is the

land-sea breeze (or bay breeze), which can prevent or reduce dispersion and contribute to enhanced secondary pollutant formation by recirculating primary urban emissions (Arya et al., 1999). This contributes to ozone exceedances and increased PM





concentrations in highly populated coastal regions (Loughner et al., 2011; Caicedo et al., 2019). Understanding this process is critical since approximately 40% of the US population lives in Coastal Shoreline Counties (NOAA, 2013). The purpose of the National Aeronautics and Space Administration (NASA) and Maryland Department of the Environment (MDE)-sponsored
Ozone Water-Land Environmental Transitions Study (OWLETS-2) was to characterize the impacts of the Chesapeake Bay and its associated meteorological phenomena on air pollution formation and transport near Baltimore (Sullivan et al., 2019). In this study, we characterize secondary aerosol formation and sources of elevated $NH_3$ measured at Hart-Miller Island, MD, a land-water transition site near Baltimore. In a companion paper, we analyze the effects of the Chesapeake Bay and elevated $NH_3$ on aerosol liquid water content and aerosol pH during OWLETS-2 (Hennigan et al., 2021).

## 2  Methods

### 2.1  Site Description

The OWLETS-2 campaign was a field study with multiple measurement platforms coordinating simultaneous water and land measurements of trace gases, aerosols, and meteorological parameters. This study presents analysis of inorganic aerosol composition and gas-phase ammonia measurements made at the Hart-Miller Island (HMI, Fig. 1) supersite. Hart-Miller Island is a
state park located on the Chesapeake Bay, approximately 22 km from downtown Baltimore.

All measurements were made in the University of Maryland, Baltimore County (UMBC) trailer located on the southeastern side of the island (39.2421°, -76.3627°). Continuous measurements were conducted from 4 June 2018 to 5 July 2018. Several gaps in data collection occurred due to periodic power outages on the island.

Comparisons of the HMI measurements were made to two nearby inland sites: Beltsville and Howard University-Beltsville
(HUB), approximately 46 km and 50 km southwest of HMI, respectively. Beltsville (39.0280°, -76.8171°) is a National Atmospheric Deposition Program Atmospheric Ammonia Site (AMoN, http://nadp.slh.wisc.edu/amon/) and was used for gas-phase $NH_3$ comparisons. HUB (39.0553°, -76.8783°) is a regulatory monitoring site maintained and operated by MDE and was used for comparisons of speciated $PM_{2.5}$ concentrations.

### 2.2  $NH_3$ Measurements

An AiRRmonia analyzer (RR Mechatronics, The Netherlands) was deployed for measurements of gas-phase ammonia (Norman et al., 2009). The analyzer was enclosed in a weatherproof aluminum box (non-temperature controlled) and placed on top of the UMBC trailer (~4 m inlet height). The AiRRmonia uses a 5 cm stainless steel inlet (I.D. = 0.2 cm) to minimize sampling artifacts associated with gas adsorption to inlet surfaces (Schmohl et al., 2001; Ellis et al., 2010; von Bobrutzki et al., 2010). The instrument samples air at a rate of 1 L min$^{-1}$, giving a gas residence time of ~0.01 s in the inlet. Ammonia in the
sampled air is absorbed through a gas-permeable polytetrafluoroethylene membrane into an acidic aqueous stripping solution with close to 100% collection efficiency (Wyers et al., 1993). NaOH is then added to raise the pH so that the collected $NH_3$ (present as $NH_4^+$) is shifted to the $NH_3$ form, facilitating transfer of $NH_3$ through another gas-permeable membrane into a



separate flow of deionized water. Dual conductivity detectors measure the difference in conductivity between the deionized water before and after this transfer, enabling quantification of sampled $NH_3$ (Norman et al., 2009; von Bobrutzki et al., 2010). The temperature-dependent conductivity was calibrated approximately daily during OWLETS-2 using aqueous $NH_4^+$ standards. The instrument's operating principles, including gas collection efficiency and lack of response to aerosol $NH_4^+$, have been characterized previously (Wyers et al., 1993; Erisman et al., 2001). The $NH_3$ measurements during OWLETS-2 were conducted with 10-minute resolution.

## 2.3 PILS-IC

Speciated inorganic $PM_{2.5}$ measurements were made with a Particle-into-Liquid Sampler (PILS) coupled to dual anion-cation ion chromatographs (IC, model 850 Dual IC, Metrohm), collectively referred to as the PILS-IC (Orsini et al., 2003). Ambient air was sampled at $16.0 \, \text{L} \, \text{min}^{-1}$ through a $PM_{2.5}$ cyclone (URG Corp.) and two annular denuders in series (URG Corp.) to remove acidic (e.g., $HNO_3$, $HCl$) and basic gases (e.g., $NH_3$). The PILS sample was split and simultaneously analyzed for concentrations of both cations ($Na^+$, $NH_4^+$, $K^+$, $Ca^{2+}$, and $Mg^{2+}$) and anions ($Cl^-$, $NO_3^-$, $SO_4^{2-}$, and $C_2O_4^{2-}$) according to (Valerino et al., 2017).

For anion analysis, the IC utilized a Metrosep A Supp 5 column ($150 \, \text{mm} \times 4.0 \, \text{mm}$) with $1.0 \, \text{mM} \, NaHCO_3$ and $3.2 \, \text{mM}$ $Na_2CO_3$ eluent at a flowrate of $0.7 \, \text{mL} \, \text{min}^{-1}$. For cation analysis, the IC utilized a Metrosep C 4 column ($150 \, \text{mm} \times 4.0$ mm) that was run with $2.5 \, \text{mM} \, HNO_3$ and $0.5 \, \text{mM} \, C_2H_2O_4$ eluent at a flowrate of $0.9 \, \text{mL} \, \text{min}^{-1}$. In the anion analysis, conductivity measurements were preceded by chemical suppression to reduce conductivity of the eluent. This configuration of the PILS-IC resulted in a limit of detection of $0.01 \, \text{µg} \, \text{m}^{-3}$ (air concentration) for each ion and a 5-min integrated sample every 20 minutes.

## 2.4 HYSPLIT Model

The Hybrid Single-Particle Lagrangian Integrated Trajectory (HYSPLIT, https://www.ready.noaa.gov/HYSPLIT.php) model was utilized to investigate source influences during OWLETS-2 (Stein et al., 2015). The model allows for air parcel trajectories, including the back-trajectory analyses used here, to infer regional source influences for the air masses sampled at HMI. High-Resolution Rapid Refresh (HRRR) 3 km meteorological data was primarily used in generating back trajectories (unless otherwise indicated). The uncertainty and limitations of the HYSPLIT model are worth noting here, especially in accounting for small-scale circulations. As a result, for all events discussed below, HYSPLIT trajectories were crosschecked with measured surface wind conditions and vertical wind fields measured from the meteorological station and a Doppler wind lidar (Leosphere Windcube 200S) deployed at HMI. Meteorological measurements of temperature, relative humidity, and surface wind speed and direction were made with the Vaisala MAWS201 Met Station at 1-minute resolution throughout the study.





## 3   Results and Discussion

### 3.1   Ammonia

During the OWLETS-2 campaign, gas-phase ammonia was characterized by unusually high concentrations. The average $NH_3$
120   concentration (3.83 $\mu g\,m^{-3}$) at HMI was more than three times greater than $NH_3$ at the Beltsville AMoN site (1.20 $\mu g\,m^{-3}$)
during the summer of 2018 (Fig. 2a), a difference that was statistically significant at the 99% confidence level (p-value < 0.01).
While average summertime $NH_3$ shows an increasing trend at Beltsville, the concentrations measured at HMI in 2018 far-
exceed recent summertime levels at Beltsville. The HMI $NH_3$ concentrations were also compared to two other AMoN sites (Fig.
1): Blackwater (~90 km from HMI; 38.4449°, -76.1112°) and Ardentsville (~110 km from HMI; 39.9231°, -77.3078°). Both of
125   these sites are located in much closer proximity to strong agricultural $NH_3$ emissions, as Blackwater is located on the Eastern
Shore of Maryland and Ardentsville is located in southeastern Pennsylvania. Their average summer $NH_3$ concentrations for
2018 were 57% (Blackwater, 1.66 $\mu g\,m^{-3}$) and 49% (Ardentsville, 1.96 $\mu g\,m^{-3}$) lower than those measured at HMI during
OWLETS-2. In a previous deployment of the AiRRmonia analyzer at the Beltsville AMoN site (unpublished work), the average
concentrations measured by the AiRRmonia over a two-week span (6 September 2016–20 September 2016) were within ~0.5
$\mu g\,m^{-3}$ of the co-located AMoN $NH_3$ measurements, suggesting that systematic measurement differences are not responsible
for the trends seen in Fig. 2a.

### 3.2   Impact on Aerosol Composition

Before investigating the sources of such high $NH_3$ concentrations, we first characterize the effects on aerosol composition,
since $NH_3$ contributes significantly to secondary aerosol formation (Seinfeld and Pandis, 2016). Note that our companion
manuscript analyzes the effects of the high $NH_3$ concentrations on aerosol liquid water and acidity (pH) (Hennigan et al.,
2021).

In the US, $SO_4^{2-}$ concentrations have decreased due to regulations and economic factors that have reduced $SO_2$ emissions
(Hand et al., 2012a). This trend is evident at HU-Beltsville (Fig. 2b), where the average summertime $SO_4^{2-}$ concentration has
decreased by more than a factor of eight since 2007 (8.23 $\mu g\,m^{-3}$ versus 0.94 $\mu g\,m^{-3}$). Note that the data shown in Fig. 2 only
include the summer months (June, July, and August) to facilitate direct comparisons to the conditions during OWLETS-2. In
contrast to the $NH_3$ results, the average $SO_4^{2-}$ concentration at HMI (1.11 $\mu g\,m^{-3}$) was quite close to the average concentration
at HUB (0.94 $\mu g\,m^{-3}$), a difference that was not significant at the 95% confidence level (p-value > 0.1). This suggests that
the high $NH_3$ concentrations at HMI did not have a large effect on particulate sulfate mass concentrations, consistent with the
understanding of $SO_4^{2-}$ thermodynamics and results from a study in the southeast US (Weber et al., 2016). The agreement in
$SO_4^{2-}$ concentrations between HMI and HU-Beltsville is unsurprising given the regional nature of $SO_2$ emissions and the low
spatial variability of $SO_4^{2-}$ levels in the eastern US (Beyersdorf et al., 2016).

While $SO_4^{2-}$ concentrations were typical for the region, the elevated $NH_3$ did affect the particulate $NH_4^+$ and $NO_3^-$ concentra-
tions. An ammonia-rich environment, defined when [Total Ammonia] > 2x[Total Sulfate], can facilitate $NH_4NO_3$ aerosol for-


mation (Seinfeld and Pandis, 2016). The system measured at HMI was ammonia-rich for approximately 25% of the OWLETS-2
campaign.

As shown in Fig. 2c, the average $NH_4^+$ concentration at HMI was more than 50% higher than the summertime average at HU-Beltsville (0.38 µg m$^{-3}$ and 0.23 µg m$^{-3}$, respectively), statistically significant at the 99% confidence level (p-value <0.01). The $NH_4^+$ concentrations at HMI were higher than any of the past five summertime averages measured at HU-Beltsville, demonstrating the influence of elevated $NH_3$. Aerosol $NO_3^-$ was also significantly elevated at HMI compared to historical and
2018 summertime levels at HU-Beltsville (Fig. 2d). The HMI average $NO_3^-$ (0.36 µg m$^{-3}$) was more than a factor of two higher than at HU-Beltsville (0.14 µg m$^{-3}$), also significant at the 99% confidence level (p-value <0.01). The $NO_3^-$ concentrations at HMI were also higher than typical summertime levels in the eastern US (Sickles II and Shadwick, 2008). The high average $NO_3^-$ concentration during OWLETS-2 was heavily influenced by several peak events where the $NO_3^-$ concentration exceeded 3 µg m$^{-3}$ (Fig. S1).

Another notable aspect of the $NO_3^-$ observed during OWLETS-2 was the diurnal profile (Fig. S2a). Formation of $NH_4NO_3$ is favored by both lower temperature and higher relative humidity (Stelson and Seinfeld, 1982). Thus, the observed midday peak in $NO_3^-$ coincided with the most unfavorable meteorological conditions (Figure xx in Hennigan et al., 2021). More typical summertime diurnal profiles of $NO_3^-$ in the eastern US exhibit a maximum around 06:00–07:00 LT and an afternoon minimum (Xu et al., 2015; Poulain et al., 2011; Weber, 2003; Wittig et al., 2004). The $NO_3^-$ diurnal profile observed during OWLETS-2
indicates the important effect of the elevated $NH_3$ during the campaign and will be discussed in more detail below.

$NH_4^+$ and $NO_3^-$ concentrations were elevated at HMI compared to the historical data (Fig. 2). In the eastern US, the PM$_{2.5}$ mass fraction of $NH_4NO_3$ peaks in the winter with a minimum in the summer (Wittig et al., 2004). Ammonium sulfate, however, is consistently one of the largest fractions of PM$_{2.5}$ mass in the eastern US (Hand et al., 2012b). For the OWLETS-2 study, the coefficient of determination ($R^2$) for the linear correlation (least squares regression analysis) between $2 \times (SO_4^{2-}$ +
$NO_3^-$ ) and $NH_4^+$ was 0.90 (Fig. 3), while the $R^2$ value for the linear relationship between $SO_4^{2-}$ and $NH_4^+$ was only 0.57 (not shown), indicating the importance of $NH_4NO_3$ and the ammonia-rich conditions during OWLETS-2.

Finally, it should be noted that $NO_3^-$ concentrations can be affected by interactions between nitric acid and sodium chloride (Seinfeld and Pandis, 2016). This can have a large impact on aerosol composition in coastal urban areas (Athanasopoulou et al., 2008). However, sodium chloride aerosols, while observed periodically in the fine mode during the campaign, were not
frequent enough or high enough in magnitude (max Na + Cl = 0.91 µg m$^{-3}$) to suggest a strong influence of primary aerosol emissions from the Chesapeake Bay.

## 3.3 Characterization of Ammonia Sources

Source locations of the high $NH_3$ events were investigated using the NOAA HYSPLIT model. Trajectories produced by the model were checked for consistency with vertical wind fields measured by Doppler wind lidar (when available). Back trajecto-
ries were initialized from HMI (50 m altitude) at the time corresponding to each $NH_3$ measurement. Air mass back trajectories were run based upon a recent study's estimate of 15 hours for the average $NH_3$ lifetime (Hauglustaine et al., 2014), though other studies have estimated much lower $NH_3$ lifetimes for point sources (Dammers et al., 2019). Figure 4 shows the ensemble





of all back trajectories for the campaign colored by the measured NH$_3$ concentration. Overall, the back trajectory origins were consistent with predominant surface winds during the campaign. Figure 4 also shows that elevated NH$_3$ concentrations were 185 not limited to trajectories originating from a single direction or region.

A subset of the trajectories in Fig. 4 was further investigated to identify the potential sources of peak NH$_3$ events. Eleven distinct events had peak NH3 concentrations above the 95$^{th}$ percentile (> 7.96 μg m$^{-3}$) for the OWLETS-2 study (Fig. 5). Back trajectories for each event are shown in Fig. 6. Altitudes of all back trajectories were below 1 km over the postulated source region.

Agriculture represents the largest source of atmospheric NH$_3$ emissions globally (Sutton et al., 2013). Agricultural NH$_3$ emissions stem from fertilizer and livestock management (Pinder et al., 2006). Previous studies have linked high livestock NH$_3$ emissions with particulate nitrate formation (Xu et al., 2019; Paulot et al., 2016; Sorooshian et al., 2008). HMI is located in relatively close proximity to several areas of concentrated livestock production (Fig. 7). To the east and southeast of HMI, the Delmarva (Delaware-Maryland-Virginia) Peninsula is a large poultry producing region. The emissions from these chicken 195 houses have been characterized by ground measurements (Siefert et al., 2004) and captured by satellite measurements (Warner et al., 2017). Additionally, there are large concentrations of dairy cows in southeastern Pennsylvania, also shown in Fig. 7. The NH$_3$ emissions in this area are some of the highest in the country for dairy cows (Pinder et al., 2004b). The back trajectories identify events #1, 2, 3, 4, 5, 6, 7, and 11 as having likely agricultural influence (Table 1), consistent with the agricultural hotspots identified here (Fig. 7).

While agricultural emissions were the likely source of many peak events, several events - including the highest NH$_3$ concentration measured during the campaign - appear to have a non-agricultural origin. Specifically, the trajectories for events #8–10 pass directly over the city of Baltimore and do not appear to intersect with any major agricultural sources in the 15 hours prior to arrival at HMI. A number of urban and industrial NH$_3$ sources are known, including motor vehicles (Bishop and Stedman, 2015; Sun et al., 2017) and fertilizer production plants (Van Damme et al., 2018). Ammonia is produced when 205 proteins are broken down under anaerobic conditions (Rittmann and McCarty, 2001), causing landfills (Sutton et al., 2000), waste composting and processing (Sutton et al., 2013; Pagans et al., 2006), and wastewater treatment plants (Reche et al., 2015; Artíñano et al., 2018) to emit significant quantities of NH$_3$, as well.

The contribution of traffic to events #8–10 was likely quite small due to a combined R$^2$ value of 0.201 between NH$_3$ and CO for the duration of the three events. NH$_3$ and CO are correlated in vehicle emissions (Kean et al., 2009; Perrino et al., 2002); 210 however, for these events, high NH$_3$ was observed without a spike in CO. We acknowledge order-of-magnitude differences in the atmospheric lifetimes of CO and NH$_3$ would alter correlations downwind from sources co-emitting both pollutants, but in this case, the species with the longer lifetime (CO) did not show an enhancement during the NH$_3$ events, suggesting mobile sources were not a significant factor here. The trajectories for events #8–10 identify a cluster of potential industrial NH$_3$ sources (Fig. 8).

The potential NH$_3$ sources for events #8–10 include the Back River Wastewater Treatment Plant (*A* in Fig. 8), the Patapsco Wastewater Treatment Plant (*B*), a W.R. Grace Chemical Production Facility (*C*), the Quarantine Road Landfill (*D*), a large-scale composting facility (*E*), and Yara Fertilizer Distributor (*F*). Of these facilities, only W.R. Grace (224.1 tons yr$^{-1}$) and





the Composting Facility ($1.03 \ \mathrm{tons \ yr^{-1}}$) are permitted to release $NH_3$ by the state of Maryland (MDE, 2019a,b). However, the non-permitted sites are likely major $NH_3$ emitters, based upon urban $NH_3$ sources previously identified in the literature. It

is likely that emissions from multiple sources collectively contributed to $NH_3$ levels during these events, however, the potential sources are too close together to distinguish or estimate individual contributions.

It is important to note that numerous back trajectories also passed over this cluster of industrial sources during times when $NH_3$ was not elevated. However, events #8–10 share characteristics that explain the high $NH_3$ levels at HMI. The peak concentration of each event occurred between 07:00–08:00 LT (Table 1). Additionally, the events had extremely low wind speeds of

$0.8 \ \mathrm{m \ s^{-1}}$, $0.7 \ \mathrm{m \ s^{-1}}$, and $0.6 \ \mathrm{m \ s^{-1}}$ respectively, all well below the campaign average of $2.9 \ \mathrm{m \ s^{-1}}$ (Table 1). These values were below the $10^{th}$ percentile of all wind speeds measured at HMI during OWLETS-2.

With such low wind speeds, transport time for the ~18 km distance between the cluster of $NH_3$ sources near downtown Baltimore and HMI would be ~7–8 hours for events #8–10, suggesting that the original emissions occurred near midnight. Low wind speeds occurring at night correspond to the most stable atmospheric conditions and to a low boundary layer, which

together enable elevated pollutant concentrations downwind of point sources (Arya et al., 1999). For example, event #10 had a peak $NH_3$ concentration ~$15.5 \ \mathrm{\mu g \ m^{-3}}$ higher than the campaign average ($3.8 \ \mathrm{\mu g \ m^{-3}}$). Under Pasquill-Gifford stability class F (nighttime, low winds), the required point source emission rate is two orders of magnitude lower ($1 \ \mathrm{g \ s^{-1}}$ vs. $172 \ \mathrm{g \ s^{-1}}$) than it is under stability class B (slight daytime solar radiation, low winds) to give the same $15.5 \ \mathrm{\mu g \ m^{-3}}$ $NH_3$ enhancement at HMI. This illustrative example shows that urban sources with moderate $NH_3$ emission rates can have a profound effect on

downwind $NH_3$ levels under the most stable atmospheric conditions.

Further, events #8–10 also occurred shortly before onsets of the Chesapeake Bay breeze. The convergence between an offshore synoptic flow and the breeze creates a period of stagnation or calm winds that favor the accumulation of pollutants (Caicedo et al., 2019; Loughner et al., 2014). Additionally, the early morning occurrence of events #8–10 also suggests a low dilution volume as the marine boundary layer is only beginning its convective regime (Stull, 1988). Combined, calm winds and

relatively shallow boundary layers are important factors contributing to the high $NH_3$ concentrations.

Natural sources of $NH_3$ can represent the dominant source of emissions in some regions (Paulot et al., 2015; Sutton et al., 2013). The Chesapeake Bay can be a source or sink for $NH_3$ in the summertime (Larsen et al., 2001); however, the emission factors reported by Larsen et al. (2001) suggest a relatively minor influence of Chesapeake Bay emissions compared to the agricultural and industrial sources discussed above.

## 3.4   Temperature Effects

Temperature plays an important role in $NH_3$ emissions from fertilizer and animal waste (Robarge et al., 2002; Pleim et al., 2013). This contributes to strong seasonal and diurnal profiles of agricultural $NH_3$ emissions (Warner et al., 2017; Zhu et al., 2015; Pinder et al., 2004a). The average diurnal profile of $NH_3$ (Fig. 9a) shows enhanced concentrations between the hours of 07:00–15:00 LT, consistent with temperature-driven emissions. Overall, $NH_3$ concentrations rose with increasing temperature

(Fig. 9b); however, there was a lot of scatter in the data ($R^2 = 0.05$ for individual (10-min) measurements of $NH_3$-temperature over the entire campaign). In the absence of strong local sources of $NH_3$ emissions, ambient observations are most impacted





by transport. At HMI, this includes diverse source sectors from different areas that correspond to various time lags in emission-to-observation. This confounds strict interpretation of temperature-dependent emission relationships.

### 3.5 Deposition Effects

Several factors likely explain why the $NH_3$ concentrations observed at HMI were more than three times higher than those observed at Beltsville (Fig. 2a). Due to differences in proximity to the city of Baltimore (Beltsville is ~45 km from downtown Baltimore) and to the Chesapeake Bay (e.g., bay breeze circulation), Beltsville was not affected by the identified industrial areas as severely as Hart-Miller Island. Further, when the wind direction may have facilitated transport from downtown Baltimore to Beltsville, the atmospheric conditions were not typically stable, giving rise to much lower downwind concentrations. However,

the trajectory analysis (not shown) indicates that Beltsville frequently received air masses that had recently passed over the agricultural areas of the Delmarva peninsula, the most common source of extreme $NH_3$ concentrations at HMI (Table 1).

The most likely explanation for the higher concentrations at HMI is a dramatic difference in the bi-directional flux of $NH_3$ during transport to each site. The path from the Delmarva Peninsula to Hart-Miller Island consists primarily of transport over water (the Chesapeake Bay), while the path for transport to Beltsville includes significantly more time and distance over land

(~40 km). The dry deposition velocity of gases is typically lower over water than over land, especially forested regions like the land surrounding the Beltsville AMoN site (Li et al., 2019). Additionally, the Chesapeake Bay can be a net source of NH3 during the summer (Larsen et al., 2001), suggesting deposition can be, on a net basis, quite small over the body of water. A recent modeling study found that reactive nitrogen undergoes much higher dry deposition to the coastline bordering the Chesapeake Bay compared to rates over water (Loughner et al., 2016). The $NH_3$ measurements at Beltsville are two-week

integrated samples, precluding a direct comparison to the HMI measurements and quantitative estimates of $NH_3$ lost to dry deposition during transport. However, dry deposition of $NH_3$ is a key uncertainty in accurate $NH_3$ predictions (Pleim et al., 2013), so future studies should examine this question directly. The dry deposition of $NH_3$ and $HNO_3$ are affected by the aerosol liquid water content and particle pH (Nenes et al., 2020), and this will be analyzed in our companion paper.

### 4 Conclusions

Semi-continuous measurements of aerosol chemical composition and gas-phase $NH_3$ were carried out during summer 2018 as part of the OWLETS-2 field campaign. The measurement site was located in close proximity to the land-water transition between the Chesapeake Bay and downtown Baltimore, MD. Sulfate levels were in line with historical trends and with concentrations measured at the same time in other locations near Baltimore. However, average aerosol $NO_3^-$ and $NH_4^+$ concentrations were significantly elevated compared to the average concentrations at a regulatory monitoring site ~45 km away. The elevated

aerosol concentrations were likely driven by gas-phase $NH_3$, as higher background levels and episodic spikes contributed to average $NH_3$ concentrations that were more than three times higher than the average levels measured at a nearby (~45 km) AMoN site.





An array of sources likely contributed to the elevated NH$_3$ concentrations. Agricultural emissions from poultry operations in the Delmarva Peninsula and dairy cattle in southeastern Pennsylvania were the most frequent sources of high NH$_3$ at HMI.
Low dry deposition velocities over the Chesapeake Bay enabled efficient transport of NH$_3$ (and likely HNO$_3$, as well) to the site. Under stable atmospheric conditions, industrial NH$_3$ emissions from a cluster of sources near downtown Baltimore also contributed to several peak NH$_3$ events, including the highest NH$_3$ concentrations measured during OWLETS-2 (> 19 µg m$^{-3}$). These observations demonstrate the complex interplay between emissions, transport, and meteorology in affecting urban aerosol chemistry. Our companion study examines the effects of the unique meteorology and high NH$_3$ concentrations
on aerosol liquid water content and acidity (pH) (Hennigan et al., 2021).

Prior studies have shown that sulfate is regional in nature in the eastern US, exhibiting low spatial variability over tens of kilometers (Beyersdorf et al., 2016). Our results are consistent with these findings, suggesting that meteorological phenomena such as the bay breeze circulation or urban heat islands (Battaglia Jr et al., 2017) do not substantially alter sulfate levels in this region. On the contrary, our results suggest strong gradients of reactive nitrogen species in the aerosol and gas-phases,
which will also produce periodic spatial gradients (including episodic spikes) in PM$_{2.5}$ mass. The relatively short duration of these events and their dependence on proximity to water suggests they may not be well captured by regulatory monitoring (or longer-term monitoring for deposition networks). These results likely have implications for atmospheric chemistry and air quality in other coastal cities, including those outside of the eastern US. Our results suggest that analyses of urban air quality may be more illuminating if the urban areas are, themselves, segregated into coastal and inland locations. For example, the
AeroCom study evaluated a number of global aerosol models using predictions in urban and remote locations as key indicators of model predictive skill (Tsigaridis et al., 2014). Our results highlight important differences in sources and processes affecting coastal versus inland urban areas that may have a determining effect on aerosol composition and concentrations.

*Code availability.* Publicly available code (https://github.com/AirChem/HYSPLITcontrol) was used to run back trajectories in bulk and scale their color by user-input data (code courtesy of Dr. Glenn Wolfe, NASA/GSFC and UMBC/JCET).

*Data availability.* Data are available at https://www-air.larc.nasa.gov/cgi-bin/ArcView/owlets.2018.

*Author contributions.* C.J.H., A.G.C., and R.D. designed the measurement plan. V.C. and R.D. carried out the meteorological and lidar measurements and analysis. N.B., M.A.B., and K.B. carried out the aerosol and gas-phase measurements and associated data processing. N.B. and K.B. performed analysis of aerosol and gas-phase data. N.B. performed the back trajectory analyses and thermodynamic equilibrium modeling. N.B. and C.J.H. wrote the primary manuscript draft, and all authors contributed to manuscript edits.

*Competing interests.* The authors declare that they have no conflict of interest.





*Acknowledgements.* A.G.C. and C.J.H. acknowledge funding from the National Science Foundation, AGS-1719252 and AGS-1719245. R.D. and V.C. acknowledge support by the National Oceanic and Atmospheric Administration – Cooperative Science Center for Earth System Sciences and Remote Sensing Technologies under the Cooperative Agreement Grant #: NA16SEC4810008. N.B. and K.B. received support through the NOAA Office of Education, Educational Partnership Program with Minority Serving Institutions (EPP/MSI).





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





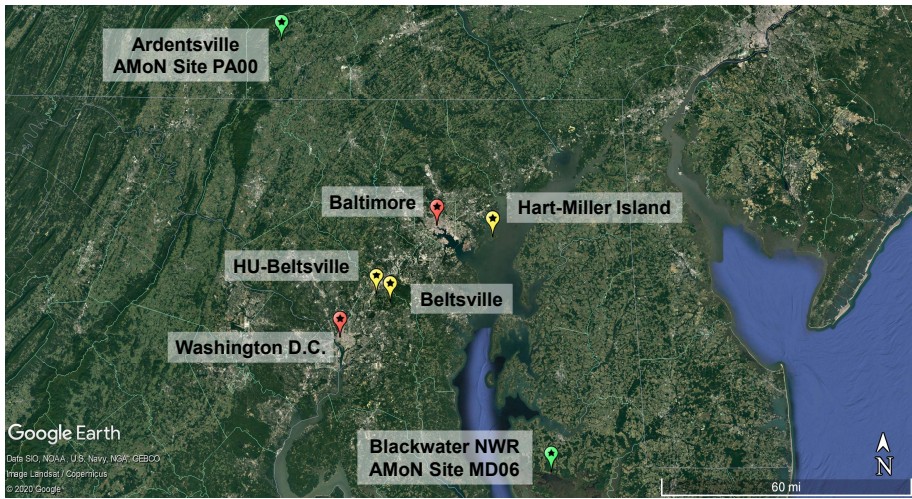

**Figure 1.** Map of relevant locations for the field campaign. Yellow stars indicate sites used for primary comparison (HMI, HU-Beltsville, Beltsville), green stars are locations used for secondary comparisons (Ardentsville, Blackwater), and red stars are labeled for spatial reference (Baltimore, Washington D.C.).

**Table 1.** Description of peak NH$_3$ events.

| Event # | Peak NH$_3$ (µg m$^{-3}$) | Time (Local) | Temperature (°C) | Wind Speed Previous 8 hour Average (m s$^{-1}$) | Source |
|---|---|---|---|---|---|
| 1 | 11.0 | 06 June 2018 23:40 | 18.9 | 2.68 | Poultry |
| 2 | 8.89 | 09 June 2018 14:45 | 24.6 | 2.09 | Dairy |
| 3 | 10.3 | 20 June 2018 13:44 | 27.7 | 2.33 | Ambiguous |
| 4 | 16.2 | 21 June 2018 19:29 | 25.3 | 2.16 | Dairy |
| 5 | 8.16 | 22 June 2018 08:19 | 22.5 | 4.50 | Poultry |
| 6 | 9.93 | 23 June 2018 14:49 | 22.7 | 3.18 | Poultry |
| 7 | 8.42 | 23 June 2018 23:39 | 23.9 | 2.08 | Poultry |
| 8 | 8.02 | 24 June 2018 07:19 | 24.5 | 0.79 | Industrial |
| 9 | 10.4 | 30 June 2018 07:15 | 26.3 | 0.67 | Industrial |
| 10 | 19.3 | 01 July 2018 07:35 | 27.8 | 0.61 | Industrial |
| 11 | 8.77 | 03 July 2018 13:42 | 31.7 | 1.77 | Dairy |

Events 2 and 3 were cut short by power outages. Event 3 trajectories suggest either poultry or dairy influence, depending on the meteorology (3 km HRRR vs. 12 km NAM).

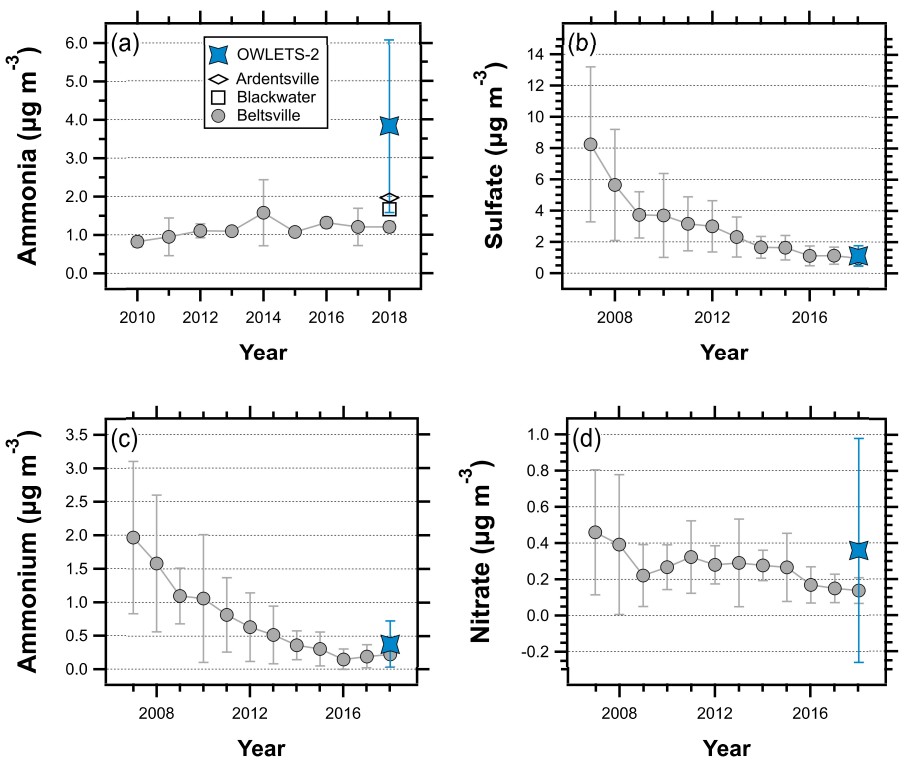

**Figure 2.** Summertime (June, July, and August) comparisons of a) $NH_3$ concentrations measured at HMI, Beltsville, Ardentsville, and Blackwater; b) $PM_{2.5}$ $SO_4^{2-}$ measured at HMI and HUB; c) $PM_{2.5}$ $NH_4^+$ measured at HMI and HUB; and d) $PM_{2.5}$ $NO_3^-$ measured at HMI and HUB. Error bars represent $\pm$ the standard deviation.

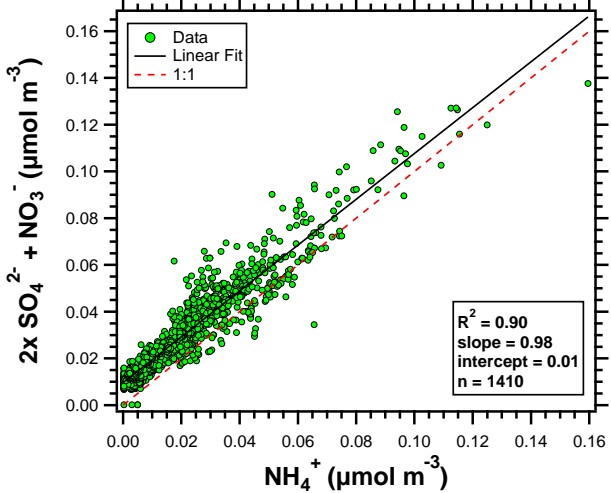

**Figure 3.** Scatter plot of $2\times$ ($SO_4^{2-}$ + $NO_3^-$) versus $NH_4^+$ (all species in molar concentrations) for the OWLETS-2 campaign.



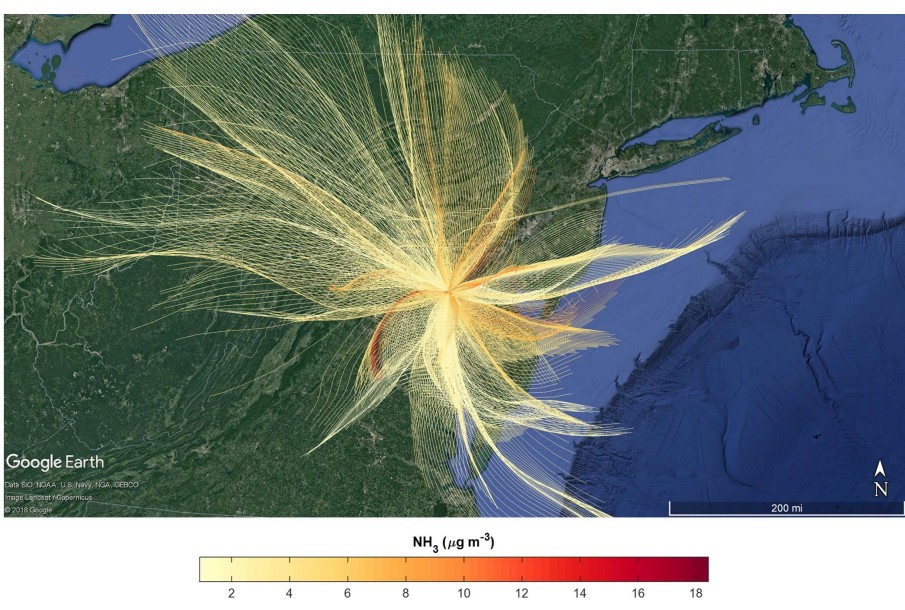

**Figure 4.** Aggregate 15-hour HYSPLIT back trajectories initialized at HMI (50 m altitude) and colored by $NH_3$ concentrations measured at HMI utilizing 12 km NAM meteorology to show regional trends. Every other back trajectory is displayed for clarity.

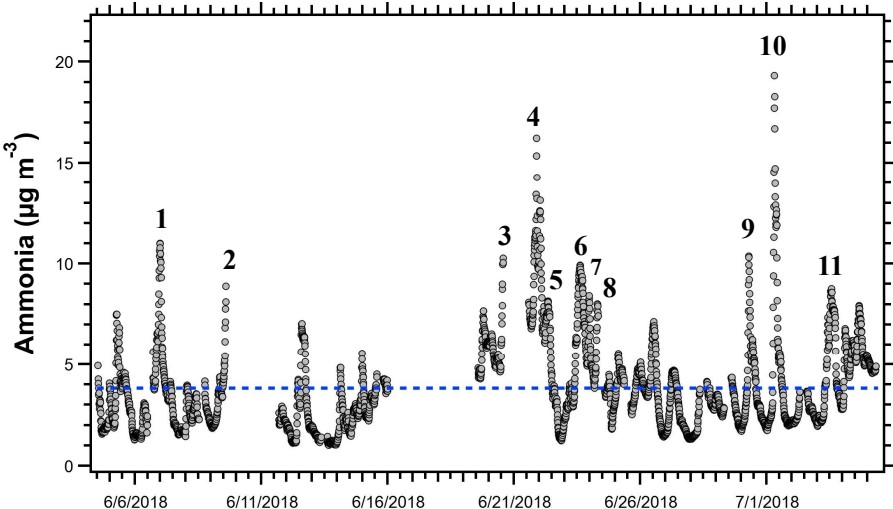

**Figure 5.** Time series of $NH_3$ measurements at HMI during OWLETS-2. The 11 $NH_3$ peak events are labeled and represent the concentrations above the 95th percentile for the entire data set (7.96 $\mu g\ m^{-3}$). Gaps in data are due to power outages that occurred on the island. Events 2 and 3 were cut off by power outages. The blue dashed line represents the campaign-average $NH_3$ concentration (3.83 $\mu g\ m^{-3}$).





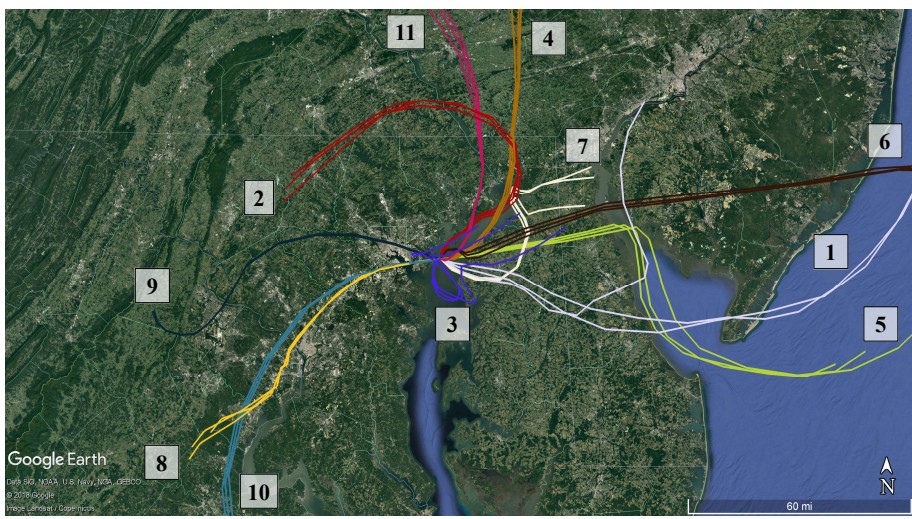

**Figure 6.** HYSPLIT back trajectories initialized at HMI using 3 km HRRR meteorology and an altitude of 50 m for the peak NH₃ events identified in Fig. 5. There are three trajectories for each event: one for the highest concentration of the event, one 10 minutes earlier, and one 10 minutes later.

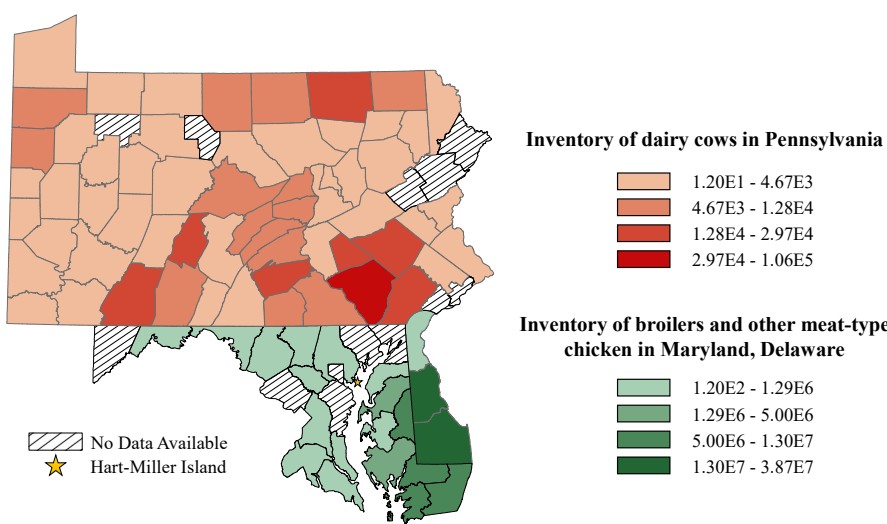

**Figure 7.** Inventories of dairy cows in Pennsylvania and broiler and other meat-type chickens in Maryland and Delaware. Data is retrieved from the 2017 Census of Agriculture (USDA, 2017).

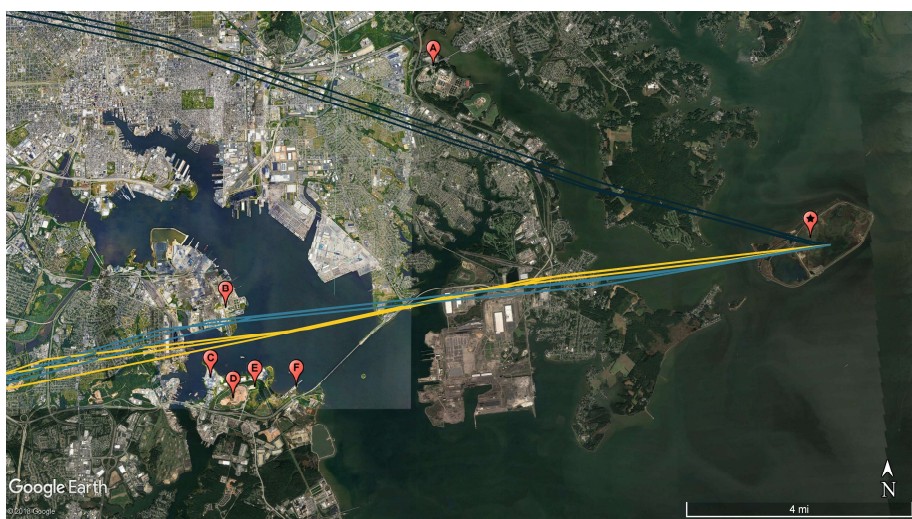

**Figure 8.** Back trajectories initialized at HMI (star) using 3 km HRRR meteorology for events #8 (yellow), 9 (dark blue), and 10 (teal). Potential industrial sources of NH$_3$ are: A) Back River Wastewater Treatment Plant; B) Patapsco Wastewater Treatment Plant; C) W.R. Grace Facility; D) Quarantine Road Landfill; E) Composting Facility; and F) Yara Fertilizer Distributor.

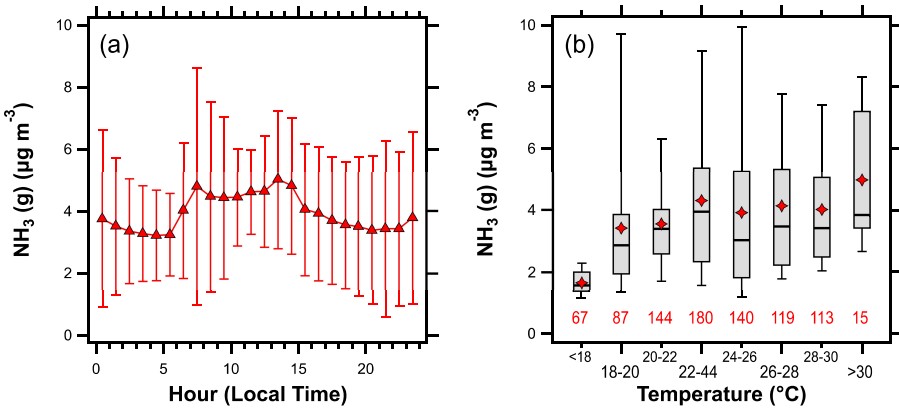

**Figure 9.** (a) The diurnal profile of NH$_3$ (error bars represent ± the standard deviation), and (b) box plot showing variations in the NH$_3$ concentration with temperature. Lines in (b) represent the 5th, 25th, 50th, 75th, and 95th percentiles, diamonds represent the mean, while numbers beneath each box represent the number of observations in each temperature interval.