# Peer review of "Urban aerosol chemistry at a land-water transition site during summer - Part 1: Impact of agricultural and industrial ammonia emissions"

_Atmospheric Chemistry and Physics, 2021_

## Author Comment (AC1)

**Response to Referee Comments**

MS no. acp-2021-365

July 28, 2021

We thank the referees for their thoughtful and thorough reviews. We have addressed their comments below, with the referee comments in bold, our responses after in plain text, and updates to the manuscript in italics.

**Response to Referee 1**

**Manuscript is very well-written, the study is a good exercise in source apportionment, that applies new concepts (land-water meteorology, stagnation events, and NH3 impacts to PM composition) to improve the hypothesized results. Only minor revisions are needed.**

1. **Figure 1 should include the NADP site ID for Beltsville site. The NH4 and NO3 data mentioned if taken from the CASTNET site is not regulatory (line 279). Only O3 is regulatory for that network. CASTNET should identified as the data source.**

   Fig. 1 has been updated to include the site ID for Beltsville. Speciated $PM_{2.5}$ data are from HU-Beltsville, which is a SLAMS (State or Local Air Monitoring Station), not CASTNET.

2. **Figure 7 mentions only dairy in PA and poultry in Maryland. Dairy is not large in Maryland, but there are some farms, since you are identifying moderate sources in downtown that are amplified due to the stagnation/low WS boundary layer events, could some of the MD dairy also be along the trajectory path?**

   To address this comment, we created Fig. S3, which is complementary to Fig. 7. Fig. S3 shows the dairy cow inventory in Maryland and Delaware and the broiler and other meat-type chicken inventory in Pennsylvania. This reveals that there are potential minor dairy sources in Maryland. Additionally, it shows that the agricultural emissions of $NH_3$ in southeastern Pennsylvania might also be contributed to, in part, by poultry. Both figures have now been formatted with a continuous color bar as opposed to discrete color bins to make these conclusions more clear.

   The text has been updated to reflect this and now reads:

   *While dairy cow populations are relatively minor in Maryland, poultry emissions could be a significant emitter in the same region of Pennsylvania where dairy cows are concentrated (Fig. S3).*

3. **Where did the CO data come from that was used to eliminate the possible traffic source?**

   The CO data used were from trace gas measurements that were co-located on Hart-Miller Island. They were conducted by the University of Maryland and NOAA and are available here: `https://www-air.larc.nasa.gov/cgi-bin/ArcView/owlets.2018?SURFACE=1`.

   The text has been updated to include this and now reads:

   *The contribution of traffic to events #8–10 was likely quite small due to a combined $R^2$ value of 0.201 between $NH_3$ and CO for the duration of the three events (based on CO measurements co-located at HMI).*

4. **NH3 needs a subscript in line 187 & 266.**

   Amended as suggested.

[Figure]

Figure S3: Inventories of dairy cows in Maryland and Delaware and broiler and other meat-type chickens in Pennsylvania. Data are from the 2017 Census of Agriculture (USDA, 2017).

[Figure]

Figure 7: Inventories of dairy cows in Pennsylvania and broiler and other meat-type chickens in Maryland and Delaware. Data are from the 2017 Census of Agriculture (USDA, 2017).

**Response to Referee 2**

This paper describes measurements of ammonia and inorganic particulate matter constituents during the 2018 OWLETS-2 campaign. Overall, it is a clear, well-written paper that provides insight into the sources of ammonia, and impacts on particle composition in a coastal, peri-urban environment. I recommend that it be accepted for publication after the authors clarify a few points, listed below.

1. The use of HYSPLIT back trajectories to interpret the data and identify the likely source of the elevated ammonia is valuable, though as the authors note, it can be challenging in near-shore environments. I have two questions related to the trajectory analysis.

    (a) **The authors indicate that the 3 km resolution HRRR data were used for trajectories (except when they weren't). Then the analysis in Figure 4 shows trajectories from the lower resolution NAM dataset. Why was the NAM data used instead of the HRRR data in some cases? It seems like it would generally be less reliable.**

    We agree with the reviewer on this point. The 12 km NAM meteorology was originally used due to data storage constraints related to the 3 km HRRR meteorology. We have updated Fig. 4, now using the 3 km HRRR meteorology and have updated the manuscript text to reflect this. We note that the updated figure is consistent with our conclusions in the original manuscript.

[Figure]

Figure 4: Aggregate 15-hour HYSPLIT back trajectories initialized at HMI (50 m altitude) and colored by NH$_3$ concentrations measured at HMI utilizing 3 km HRRR meteorology. Every other back trajectory is displayed for clarity.

    (b) **On lines 111-115, the authors note that they compared the trajectory analysis to the local observations, but they do not indicate what, if anything, was done in the analysis if the observations and modelled trajectory disagreed.**

    In order to crosscheck the back trajectories, comparisons were made between the wind direction

predicted by HYSPLIT and the wind direction measured by Doppler wind lidar when it was available and surface meteorology when wind lidar data were unavailable. Events #3–11 all had wind lidar data available and the wind direction at 50 m matched the wind direction predicted by HYSPLIT. For Events #1–2, it was necessary to use surface wind conditions to crosscheck. Event #1 matched HYSPLIT, while there was a slight discrepancy with Event #2 (HYSPLIT predicts wind from the NNE, surface wind direction was measured to be from the E). The text has been updated to note this discrepancy and now reads as follows at the beginning of Section 3.3:

*Trajectories produced by the model were checked for consistency with vertical wind fields measured by Doppler wind lidar (when available, with the one discrepancy noted in Table 1).*

The footnote of Table 1 now reads:

*Event 2 (one of the two events where wind lidar data were not available for crosschecking the back trajectory) is the only event with a discrepancy between HYSPLIT (predicts wind direction coming from NNE) and observations (measured surface wind direction coming from E).*

2. **A major factor that is emphasized in the analysis is that the gas phase ammonia concentrations at the Hart Miller Island (HMI) site are much higher than the three closest AMoN sites. Figure 2, and the corresponding text shows summertime (June, July, August) annual average concentrations from the AMoN sites in comparison with the one month of data from HMI. Are the authors confident that the differences are not, in part, caused by a difference in averaging period? In many locations, ammonia measurements are much higher in late spring than in the middle of summer. Why not average the AMoN data just for the time period that overlaps with the OWLETS-2 campaign?**

The measurements from AMoN were averaged over the entire summertime (and thus the speciated $PM_{2.5}$ from HU-Beltsville for consistency) because AMoN measurements of $NH_3$ over that time period are 2-week average samples (with some missing samples). This makes comparisons from year-to-year quite challenging for specific dates, as with the OWLETS-2 campaign. The entire summer was used instead to facilitate greater sample numbers in the statistical analysis. However, for thoroughness, we recreated Fig. 2 (see Fig. 2R below) with the averages over any 2-week period that intersected OWLETS-2 to show that it does not make a significant difference. Therefore, we will keep the original analysis and Fig. 2.

3. **Section 3.3 – in principle, the sum of gas phase ammonia and particle phase ammonium should be a better conserved tracer reflecting high emissions of ammonia. It appears that, in general, gas phase ammonia is present in much higher levels and particle phase ammonium, so the peaks identified in Figure 5 would also represent the main episodes of high total ammonia, but can the authors confirm?**

We agree that total ammonia would be a better tracer for ammonia emissions. Gas-phase ammonia was used as a tracer for ammonia emissions instead of total ammonia due to better data coverage (e.g., missing particle-phase ammonium from 6/24 – 6/29). However, when both ammonium and ammonia were measured, total ammonia correlated strongly with gas-phase ammonia (R = 0.9927) and thus works well as an indicator of total ammonia. The text has been updated to clarify this and now reads:

*When both particle-phase and gas-phase ammonia were measured, total ammonia and gas-phase ammonia correlated strongly (R = 0.9927), showing that gas-phase ammonia is an effective tracer for total ammonia for this campaign.*

4. **Lines 227-235 I found the example provided about the stability classes a bit too narrow and specific to apply generally to the dataset. I believe that the authors are talking specifically about the impact of dispersion on the surface concentration enhancement? It might be good to clarify.**

We believe this example to be necessary to show the importance of atmospheric stability and the resulting dispersion that could occur during transport from the proposed urban ammonia sources to Hart-Miller Island. However, we agree that it is important to emphasize that this is an illustrative

example that should not be generalized to the entire data set. As a result, we have edited the text so that it now reads as follows:

*This illustrative example shows the potential for atmospheric stability to impact downwind surface concentration enhancement. Specifically, for this study, urban sources with moderate $NH_3$ emissions can have a profound effect on downwind $NH_3$ levels under the most stable atmospheric conditions.*

5. **I suggest including the dates of the measurement campaign in the abstract.**

Amended as suggested.

6. **Line 162 – Figure XX needs to be corrected to the actual number.**

Amended as suggested.

[Figure]

Figure 2R: Summertime (June, July, and August) and OWLETS-2 comparisons of a) $NH_3$ concentrations measured at HMI, Beltsville, Arendtsville, and Blackwater; b) $PM_{2.5}$ $SO_4^{2-}$ measured at HMI and HUB; c) $PM_{2.5}$ $NH_4^+$ measured at HMI and HUB; and d) $PM_{2.5}$ $NO_3^-$ measured at HMI and HUB. Error bars represent $\pm$ the standard deviation.

**References**

USDA: Census of Agriculture, Volume 1, Chapter 2: County Level Data, Tech. rep., 2017.